# Newcastle Disease Virus (NDV) Oncolytic Activity in Human Glioma Tumors Is Dependent on CDKN2A-Type I IFN Gene Cluster Codeletion

**DOI:** 10.3390/cells9061405

**Published:** 2020-06-05

**Authors:** Noemi García-Romero, Irina Palacín-Aliana, Susana Esteban-Rubio, Rodrigo Madurga, Sergio Rius-Rocabert, Josefa Carrión-Navarro, Jesús Presa, Sara Cuadrado-Castano, Pilar Sánchez-Gómez, Adolfo García-Sastre, Estanislao Nistal-Villan, Angel Ayuso-Sacido

**Affiliations:** 1Faculty of Experimental Sciences, Universidad Francisco de Vitoria, 28223 Madrid, Spain; noemigromero@gmail.com (N.G.-R.); rmadurga@fundacionhm.com (R.M.); peps86@gmail.com (J.C.-N.); 2Brain tumour laboratory, Fundación Vithas, Grupo Hospitales Vithas, 28043 Madrid, Spain; 3Atrys Health, 08025 Barcelona, Spain; ipalacin@atryshealth.com; 4Fundación de Investigación HM Hospitales, HM Hospitales, 28015 Madrid, Spain; 5Facultad de Medicina, Instituto de Medicina Molecular Aplicada (IMMA), Universidad San Pablo-CEU, 28668 Madrid, Spain; susana.rubio91@gmail.com (S.E.-R.); ser.rius.ce@ceindo.ceu.es (S.R.-R.); 6Microbiology Section, Dpto. CC, Farmacéuticas y de la Salud, Facultad de Farmacia, Universidad San Pablo-CEU, 28668 Madrid, Spain; 7CEMBIO (Centre for Metabolomics and Bioanalysis), Facultad de Farmacia, Universidad San Pablo-CEU, 28668 Madrid, Spain; 8Independent researcher, 28003 Madrid, Spain; jesus_l_presa@yahoo.es; 9Department of Microbiology, Global Health and Emerging Pathogens Institute, Icahn School of Medicine at Mount Sinai, New York, NY 10029, USA; sara.cuadrado@mssm.edu (S.C.-C.); adolfo.garcia-sastre@mssm.edu (A.G.-S.); 10Neurooncology Unit, Instituto de Salud Carlos III-UFIEC, 28220 Madrid, Spain; psanchez@isciii.es; 11Department of Medicine, Division of Infectious Disease, Icahn School of Medicine at Mount Sinai, New York, NY 10029, USA; 12The Tisch Cancer Institute, Icahn School of Medicine at Mount Sinai, New York, NY 10029, USA

**Keywords:** glioblastoma, oncolytic virotherapy, Newcastle disease virus (NDV), Interferon I

## Abstract

Glioblastoma (GBM) is the most aggressive and frequent primary brain tumor in adults with a median overall survival of 15 months. Tumor recurrence and poor prognosis are related to cancer stem cells (CSCs), which drive resistance to therapies. A common characteristic in GBM is *CDKN2A* gene loss, located close to the cluster of *type I IFN* genes at Ch9p21. Newcastle disease virus (NDV) is an avian paramyxovirus with oncolytic and immunostimulatory properties that has been proposed for the treatment of GBM. We have analyzed the *CDKN2A-IFN I* gene cluster in 1018 glioma tumors and evaluated the NDV oncolytic effect in six GBM CSCs ex vivo and in a mouse model. Our results indicate that more than 50% of GBM patients have some *IFN* deletion. Moreover, GBM susceptibility to NDV is dependent on the loss of the *type I IFN*. Infection of GBM with an NDV-expressing influenza virus NS1 protein can overcome the resistance to oncolysis by NDV of type I-competent cells. These results highlight the potential of using NDV vectors in antitumor therapies.

## 1. Introduction

Glioblastoma (GBM) is the most devastating and least treatable brain tumor, with an incidence of 3.2 cases per 100,000 inhabitants [1]. GBM cells grow quickly and infiltrate the brain parenchyma [2]. The morphology and location of the tumor, as well as the clinical status and age of the patients determine the therapeutic treatment and its efficacy. After surgery, the standard GBM treatment continues with the administration of temozolomide (TMZ) concomitant with radiotherapy [3]. Despite advances in medical image and novel therapies, the median overall survival is 15 months [4], suggesting that other alternatives are needed to find more effective treatment strategies for this heterogenous disease. In this sense, several molecular and genetic alterations have been observed that contribute to tumorigenesis and treatment resistance [5,6]. Some of the most relevant mutations in gliomas are the homozygous and hemizygous deletion at Ch9p21 [7], which involves the genetic loss of *CDKN2A* (p16^INK4^) [8,9]. Interestingly, the *type I Interferon (IFN)* gene cluster share the same chromosome location and the codeletion of both clusters has been further reported [10,11,12,13]. *IFN* deletion implies the loss of an efficient antiviral response. The absence of the *IFN* cluster might make these tumors more vulnerable to virus oncolytic therapies.

Newcastle disease virus (NDV) has been proposed as an oncolytic agent for the treatment of different tumors, including GBM [14,15]. It is an avian paramyxovirus whose oncolytic potential derives in part from its capacity to preferentially replicate in tumor cells [16,17,18] and its inherent ability to trigger *type I IFN* response in humans, mediating the expression of numerous interferon-stimulated genes (ISGs) [19,20]. Many ISGs generate an antiviral state in the target cells blocking the replication of the virus [21]. Different studies have shown that deficiencies in the *IFN* induction and signaling result in the favorable replication of NDV in tumor cells [22,23]. The oncolytic properties of NDV have been improved using several strategies, including the genetic modification and the insertion of different genes with immunostimulating or immunomodulating properties [24]. 

In the present work, we have evaluated the genetic loss of the *type I IFN* locus in 1018 tumors whose copy number alteration (CNA) data are available in the TCGA database [25]. The analysis shows a correlation between *CDKN2A* deletion and the partial or total loss of the *type I IFN* gene cluster. GBM CSCs from different patients were evaluated for the presence of *CDKN2A* and *IFNβ1* genes. We also evaluated whether the *type I IFN* gene cluster determines the ability of NDV virus to replicate and induce CSCs death. Interestingly, the poor NDV oncolytic activity on type I-IFN-competent GBM could be overcome using recombinant NDV expressing a *type I IFN* antagonistic protein such as the non-structural 1 (NS1) protein of the influenza virus. That viral protein is responsible for inhibiting host immune responses [26,27]. These results highlight the importance of evaluating *IFN* competence of tumor cells for the choice of a therapy based on virus oncolysis.

## 2. Materials and Methods

### 2.1. Bioinformatic Analysis of CNA

DNA copy number of 1018 glioma (astrocytoma *n* = 169, oligoastrocytoma *n* = 113, oligodendroglioma *n* = 172, and GBM *n* = 564) samples were obtained from the work of Ceccarelli et al. [25]. The amplification/deletion state of a set of *IFN* genes and *CDKN2A*, all of them located at chromosome 9p21.3, were rank normalized and then hierarchically clustered with centroid linkage and the usual metric.

### 2.2. Cells

Vero (ATCC^®^ CCL-81^™^, Manassas, VA, USA) and Hep2 (ATCC^®^ CCL-23^™^, Manassas, VA, USA) were cultured in DMEM supplemented with 10% FBS (Gibco™) and 1% of penicillin/streptomycin (Gibco™, Waltham, MA, USA). 

Primary GBM stem cells called GBM18, GBM27, GBM38, GBM123, GBM128B, and GBM128D were obtained from primary solid tumors from patients at the Hospital Universitario La Fe in Valencia, Spain. The samples were obtained under the approval of the hospital ethics committee and the patients or their legal representatives’ agreement. All GBM cells were obtained and grown in M21 media containing DMEM/F-12 (Gibco, 11039), non-essential amino acids (10 mM; Gibco, 11140), Hepes (1 M; Gibco, 15630), D-glucose (45%; Sigma, St. Louis, MO, USA; G8769), BSA-F5 (7.5%; Gibco, 15260), sodium pyruvate (100 mM; Gibco, 11360), L-glutamine (200 mM; Gibco, 25030), antibiotic-antimycotic (100×; Gibco, 15240), N2 supplement (100×; Gibco, 17502), hydrocortisone (1μg/μL; Sigma, H0135), tri-iodothyronine (100 μg/mL; Sigma, T5516), EGF (25 ng/μL; Sigma, E9644), bFGF (25 ng/μL; Sigma, F0291), and heparin (1 μg/μL; Sigma, H3393), following the same process previously described [28]. Cells were maintained in an incubator at 37 ^°^C with 5% of CO_2_ and 90% humidity. Lipofectamine™ 3000 Reagent (Thermofisher Scientific, Waltham, MA, USA) was used to transfect CSCs with luciferase-expressing plasmids (pLV[Exp]-Hygro-EF1A > Luciferase) following manufacture’s procedure.

### 2.3. gDNA Isolation and PCR

Genomic DNA (gDNA) was isolated using DNeasy Blood and Tissue Kit (QIAGEN) following the manufacturer’s recommendations. All PCRs for the amplification of *CDKN2A*, *IFNβ1* and *β-actin* were performed following the protocol of the Paq5000 enzyme (Agilent Technologies, Las Rozas de Madrid, Madrid, Spain). 

### 2.4. Viruses

Recombinant NDV-LaSota-F3aa-GFP (rNDV) was rescued and grown as described previously [29]. Recombinant viruses NDV-LaSota -GFP (rNDV-GFP) and rNDV-LaSota-NS1 viral stocks were generated as previously described [30] and stored at −80 °C. Schematic representation of the viruses genome is provided in Appendix A. Stocks’ titration was performed by indirect immunofluorescence using a rabbit polyclonal serum to NDV [31] on Hep2 infected-cells, 48 h after viral inoculation. TCID50 was calculated by the Reed-Muënch method [32]. The multiplicity of infection (MOI), which refers to the number of infectious particles per cell, has been indicated in each experiment. 

### 2.5. Infectivity Assay

10,000 GBM cells were plated on a 96-well plate and infected with rNDV-GFP at 0.01 (low) or 1 (high) MOI. To avoid artifacts during cell fixation, 20 µL of DRAQ5 fluorescent probe (Thermofisher) was added to stain cellular DNA after 24–72 h of incubation and 15 min later different fluorescence images were taken.

### 2.6. Replication Assay

Viral-growth kinetics were performed in 6-well plates containing 80,000 cells per well. Cells were inoculated with a MOI of 0.01 in Opti-MEM^®^ media (Gibco) for 1 h at room temperature (RT). After inoculum removal, cells were maintained in M21 media. Supernatants were collected every 24 h for a total extent of one week and titrated by immunofluorescence as described before. Infected cells were harvested at the given time points, to perform gene expression analysis.

### 2.7. Cell Viability Assays

96-well plates were used to plate 10,000 cells/well in 100 µL of M21. Cells were infected at the indicated MOI using Opti-MEM^®^ media (Gibco). After one hour of incubation at RT, the inoculum was removed, and the cells were maintained in M21 media. 

Cell viability was assessed at the indicated time points for an extent of 7 days by using MTS reagent (Promega, Madison, WI, USA) following manufacturer indications. Absorbance was recorded at 490 nm and 630 nm using a spectrophotometer plate reader. Values were calculated in reference to the viability of mock-infected cells (negative control).
Cell viability (%) = [Absorbance infected-sample/Absorbance mock-infected sample] × 100

### 2.8. Gene Expression

RNA was extracted using RNeasy Mini Kit (Qiagen, Germantown, MD, USA) following the manufacturer instructions. After RNA isolation, DNAse treatment was performed using DNAse I (Qiagen) following manufacturer’s instructions. RNA concentration and purity assessment were performed by using a Nanodrop (Thermofisher), then samples were stored at −80 °C. cDNA was generated from 1 µg of purified RNA by using high-capacity cDNA reverse transcription Kit (Applied Biosystems, Carlsbad, CA, USA) following manufacturer instructions. Quantitative PCR was performed using SYBR Premix Ex Taq (Takara, Kusatsu, Shiga, Japan) following manufacturer instructions in a 7900HT fast real-time PCR system (Thermofisher) using the primers needed to quantify the expression of the genes under analysis. β-actin and GAPDH were used as housekeeping genes. Primer couples used to quantify the expression of each gene were as follows: *β-actin* Fw: CATCCCCCAAAGTTCACAAT and *β-actin* Rv: ATGGCAAGGGACTTCCTGTA; *GAPDH* Fw: TCCTCCACCTTTGACGCTG; *GAPDH* Rv: ACCACCCTGTTGCTGTAGCC; *IFN-α* Fw: AGAATCTCTCHTTYCTCCTG; *IFN-α* Rv: TTCTGCTCTGACAACCTCC; *IFN-β* Fw: CGACACTGTTCGTGTTGTCA; *IFN-β* Rv: GAAGCACAACAGGAGAGCAA; *ISG54* Fw: ATGTGCAACCTACTGGCCT; *ISG54* Rv: TGAGAGTCGGCCCATGTGA; *MDA5* Fw: CAACATGGGCAGTGATTCAGG; *MDA5* Rv: TGGGCAACTTCCATTTGGTAAG; *MXA* Fw: GTTTCCGAAGTGGACATCGCA; *MXA* Rv: GAAGGGCAACTCCTGACAGT; *NDV* Fw: ATTGCWGTYAGYGAGGATGC; *NDV* Rv: GTCTCRTAHGCWGTTCATGGC; *OAS1* Fw: GATCTCAGAAATACCCCAGCCA; *OAS1* Rv: AGCTACCTCGGAAGCACCTT; *RIG-I* Fw: TGGTTCCGTGGCTTTTTGGATGC; *RIG-I* Rv: TTCCACAACCTGTAGGAGCACATA. Relative quantification of the gene expression changes was analyzed according to the 2^−ΔΔCT^ method.

### 2.9. IFN Treatment

CSCs were treated with 500 U/mL of recombinant *IFN-β* (PBL Assay Science, Piscataway, NJ, USA). Pictures were taken on experimental days 0, 2, and 7. Sphere size was measured with CellProfiler 3.1.9 (Broad Institute, Cambridge, MA, USA), objects higher than 1500 pixels were used for the analysis. Total RNA was isolated at 24 h for further gene expression analysis. 

### 2.10. Orthotopic GBM Xenotransplants 

Total of 4 × 10^5^ GBM 18 and GBM 27 luciferase reporter-cells were stereotactically injected into the striatum of the right hemisphere (0 mm anterior and 2.5 mm lateral to the bregma; 3.5 mm intraparenchymal) of 9-week-old NUDE mice brain (Charles River Laboratories, Wilmington, MA, USA). Tumor growth was monitored using the IVIS Lumina III system (Perkin Elmer, Waltham, MA, USA) after intraperitoneal injection of D-luciferin (30 mg/mL) (Gold Biotechnology, Ashby, MO, USA).

When tumors reached a standard size, mice received a single intratumoral dose of 5 × 10^4^ iu/mL of rNDV-GFP virus or the corresponding HBSS controls. Images were analyzed using Image J software 1.48q (National Institute of Mental Health, Bethesda, MD, USA). Mice were euthanized when they presented neurological symptoms or a significant weight loss [≥20% of starting body weight]. 

### 2.11. Evaluation of NDV Toxicity in Mice

Total of ten Nude and five CD1 mice received a single intrathecal dose of 5 × 10^4^ IU/mL rNDV-GFP virus as described before. Animals were euthanized when displayed distress symptoms or experienced 20% of body weight loss.

### 2.12. Statistical Analysis

Statistical analysis was performed using a 2-tailed Student t test and two-way ANOVA. Data are presented as means ± standard deviation and were calculated using the software package GraphPad Prism v. 5.0. (GraphPad Software, San Diego, CA, USA). Statistical values of *p* < 0.05 were considered significant.

## 3. Results

### 3.1. Codeletion of CDKN2A and the Type I- IFN Cluster Is Very Common in Glioma

In order to study the prevalence of *CDKN2A* in brain tumors, we performed an in silico analysis of 1018 glioma tumors. Our results revealed that GBMs have the highest occurrence of *CDKN2A* gene deletion among gliomas (Figure 1a). 76% of the GBM tumors included in this analysis (*n* = 428) were *CDKN2A* deficient, with homozygous and heterozygous deletions being present on the 57% (*n* = 322) and 19% (*n* = 106) of them, respectively. 

Because of the proximity of the *type I IFN* gene cluster to *CDKN2A* at 9p21 (Figure 1b), we decided to investigate whether the coverage of the deletion further included this set of genes (Figure 1c,d). Gene clustering pointed out that *CDKN2A* gene state is coordinate with *type I IFN* genes and our analysis shows that most of the patients with heterodeletion in the gene *CDKN2A* presented also partial deletion of the *type I IFN* cluster. Moreover, the *CDKN2A* homozygous deletion in GBM and astrocytoma entail either partial or complete codeletion of the *type I IFN* cluster in more than 50% of these tumors (Figure 1d). In contrast, the incidence of *type I IFN* deficiency for oligoastrocytoma and oligodendroglioma groups remained lower (Figure 1e). 

In light of the results obtained in our in silico analysis, we decided to test if this phenomenon prevailed in a set of primary GBM CSCs derived from primary tumors of glioma patients. Considering that *IFNβ1*, the gene that codifies for *IFN-β*, is the one located the farthest from *CDKN2A*, we assumed that the absence of both genes implies the loss of the entire *type I IFN* cluster. The genetic analysis of these CSCs showed that three of them (GBM27, GBM128B, and GBM128D) presented a codeletion of the *CDKN2A* and *IFNβ1* genes (type I non-competent CSCs). On the contrary, GBM18, GBM38, and GBM123 (type I-competent CSCs), displayed a preservation of both genes (Figure 1f or Appendix A). Both sets of CSCs were used to investigate the role of *CDKN2A-type-I IFN* phenotypes.

### 3.2. Loss of the IFN Cluster Conditionates NDV Replication and Redefines the Antiviral Response of GBM Cells

In order to study the ability of NDV to replicate in GBM CSCs, we infected the cells at a low MOI (0.01) and monitored the viral growth kinetics as well as the induction of the transcription of *IFN-β* and *MX1*, a gene highly expressed exclusively under *IFN* detection, over a time course of 120 h. Consistent with previous data [23], NDV demonstrated limited replication capacity in *type-I-IFN*-competent CSCs. In contrast, NDV had higher titers and/or more prolonged replication in GBM27, GBM128B, and GBM128D, which correlated with the induction of *IFN-β* and *MX1* gene expression (Figure 2a). 

To characterize the *type I IFN* response in the different GBM CSCs, we infected them at a high MOI (4) and analyzed the expression of different interferon stimulated genes (ISGs) at 24 h after infection. As shown in Figure 2b, we could differentiate GBM cells into two clusters, according to their ability to express *IFN-α*, *IFN-β,* and different *ISGs*. As expected, both *IFN-α* and *IFN-β* genes were induced in type I-competent CSCs upon NDV infection. In comparison, GBM27, GBM128B, and GBM128D did not present a strong response to NDV infection. It can be noticed that GBM128D presents a small but significant stimulation of *IFN-α* and *IFN-β* genes (Figure 2b). 

As expected, the analysis of the expression of relevant ISGs involved in the cellular antiviral response to NDV, such as *RIG-I*, *MDA5*, *STAT1*, *Mx1*, *OAS1*, *ISG54* or *ISG15*, shows an upregulation of these genes on GBM18, GBM38, and GBM123 cells. In contrast, the level of induction of the same genes in the type I- non-competent CSCs is considerably lower upon NDV infection.

We further investigate other important parameters of the response of cancer cells to NDV, including upregulation of MHC molecules and inflammatory cytokines. Surprisingly, MHC-I mRNA expression could not be detected in GBM18 and low expression values were observed in GBM123 cells. GBM38 showed a basal expression of MHC-I but this gene was not stimulated upon NDV infection. In comparison, MHC-I expression was significantly stimulated upon NDV infection in GBM27, GBM128B, and GBM128D. Nuclear factor kappa B (NF-κB)-dependent pro-inflammatory genes such as interleukin-6 (IL-6) and tumor necrosis factor (TNF-α) were also analyzed. All cell types were able to upregulate the expression of these genes upon NDV infection, except for GBM27. NDV infection can lead to apoptosis through the expression of *FAS* or TNF-related apoptosis inducing ligand (*TRAIL*) genes. The expression analysis of the cells showed that all cells were able to stimulate the expression of both genes with the exception of GBM27, that was unable to induce *FAS* expression, and GBM128B that did not stimulate *TRAIL* expression upon infection. However, GBM128D induction levels did not reach the basal level of GBM128B (Appendix A).

### 3.3. Type-I Deficient GBM CSCs Are More Susceptible to NDV-Derived Oncolysis 

A major feature of NDV’s oncolytic activity is its capacity to induce cell death in tumor cells. In order to determine if the loss of the *IFN* cluster could affect the susceptibility of GBM cells to viral-induced cytopathic effect (CPE), we performed a MTS cell viability assay on CSCs infected at different MOIs. As observed in Figure 3a, all the GBM cells were susceptible to be infected by rNDV-GFP virus. However, a clear dependence on MOI was noted along all groups, in reference to CPE or loss of viability (Figure 3b). 

GBM18 and GBM123 (type I-competent) CSCs displayed the highest level of resistance to NDV-induced cell death, showing significant loss of cell viability (≥50%) at a MOI higher than 1. In the case of GBM38, this effect was reached with a MOI of 0.1 (Figure 3b). In comparison, a MOI of 0.01 or lower is sufficient to reduce the cell viability below 50% in *type-I-IFN* non-competent CSCs. Noteworthy, GBM27 cells showed the highest susceptibility to NDV-induced cell death, with a reduction on cell viability higher than 50% at the lowest MOI used in this study (MOI of 0.001) (Figure 3b).

To better characterize the CPE induced by NDV, GBM CSCs were infected at low and high MOIs (0.01 and 1 respectively) and loss of viability was monitored every 24 h, over a time course of 120 h. Type-I-IFN-competent CSCs displayed a significant drop in viability (≥50%) at 48 h post-infection at a MOI of 1 (Figure 3c). However, and confirming our previous results, this type I-competent CSCs presented high resistance to virus when they were infected at a low MOI (Appendix A). In contrast, GBM27, GBM128B, and GBM128D non-competent cells were more susceptible to the virus infection at a MOI of 0.01 than the *IFN* competent cells over the extent of the study. This effect was observed after 24 h of NDV infection in GBM128B and GBM128D. Instead, GBM27 presented a delayed susceptibility to the virus that is significative only after 48 h of NDV infection.

In order to confirm the lytic effect of NDV in vivo, we choose GBM18 and GBM27 luciferase-expressing cells, *as type-I-IFN* competent and non-competent CSCs respectively, to develop GBM xenotransplant tumors in mice. Tumor growth was monitored using an IVIS luminometer using the luciferase intensity of these tumors as a surrogate measure of the tumor size. When the tumors reached an average radiance of 6.77 × 10^3^ [p/s/cm^2^/sr], they were injected with 5 × 10^4^ IU/mL of rNDV-GFP. Seven days after intratumoral NDV inoculation, we observed a significant decrease in the luciferase expression only in the *IFN* deficient GBM27 xenotransplanted tumors (*p* < 0.001) (Figure 3d). 

### 3.4. GBM Cells Are Able to Respond to Recombinant IFN-β Treatment 

The classification of our GBM CSCs as *type-I IFN* competent or non-competent was based on the presence of *IFNB* deletion and, therefore, in their individual capacity to produce *IFN-β*. We have already described the autocrine effect of the *IFN-β* produced by these cells (Figure 1; Supplementary Figure 1). However, whether or not these cells could respond differently to a stronger exogenous *IFN-β* stimulation needed to be stated. To investigate if the CSCs were able to respond to exogenous interferon, GBM cells were treated with 500 U/mL of recombinant *IFN-β*. As shown in Figure 4a, no morphological changes were noted between treated and untreated samples. However, an anti-proliferative effect of the addition of exogenus *IFN-β* was noticed for all the cell lines studied. This biological effect was statistically significant in GBM18 (*p* < 0.05) and GBM27 (*p* < 0.01). When the sphere diameters were measured, we observed a dramatical decrease in GBM27 (*p* < 0.01) and in the relative expression of stemness-associated genes (*BMI1* and *CD44*). The addition of *recombinant IFN-β* also triggered a significant increase in the expression of *MX1*, *OAS1,* and *ISG15* genes (Figure 4b). No dramatical changes were observed in the expression of pro-inflammatory cytokine IL-6 or chemokine CXCL1, except for the cell lines GBM123 and GBM27, in which CXCL1 levels only increased after addition of *IFN-β*.

### 3.5. NDV Lytic Activity Could Be Restored in IFN-Competent GBM Cells by Coexpressing a Type I IFN Antagonist

Based on our previous data, NDV infection has a stronger lytic effect and therapeutic potential on *IFN*-deficient GBM. Therefore, we decided to test whether a recombinant NDV (rNDV-NS1) bearing the NS1 gene from influenza A virus, a strong antagonist of the *IFN* response, could exert higher oncolytic activity in these *IFN*-competent CSCs. The insertion of the NS1 into the NDV genome has previously probed to enhance the viral replication and lytic capacity of NDV in other human cancer cells [31]. 

For this purpose, we used GBM18 and GBM38 as *IFN* competent CSCs and GBM27 and GBM128D as deficient CSCs (Figure 5 and Appendix A). As shown in Figure 5a, rNDV-NS1 viral titer was 1000-fold higher than rNDV-GFP virus in GBM18 and were more effective at high MOI reducing the number of viable cells below 20% at 120 h. The rNDV-NS1 decreased the viability in GBM38 drastically compared with rNDV-GFP at both MOIs. However, the growth kinetics of rNDV-GFP and rNDV-NS1 viruses did not change in GBM27 and GBM128D and similar viability effects were observed when these CSCs were infected with MOI 0.01 and 1 (Figure 5b or Appendix A). We further confirmed the *IFN-β* inhibition by mRNA expression and observed that in competent CSCs model GBM18 the expression decreased drastically when rNDV-NS1 was used in the infection (*p* < 0.001). As expected, no changes were observed in the non-competent CSCs model.

## 4. Discussion

Despite the development of new drugs, GBM remains the most lethal brain tumor, mainly due to its resistance to conventional treatments, high heterogeneity and the presence of CSCs [33]. Although GBMs can be classified based on histology and molecular characteristics [34,35,36], there are no specific genetic biomarkers or signatures for treatment stratification and the survival rates have remained unchanged during the past years [37]. 

Our group has isolated and characterized several primary cultures of CSCs which nowadays are featured as the best GBM *in vitro* models [28]. In the present work we propose that NDV oncolytic therapy could be a good approach for the treatment of a specific subgroup of primary brain tumors. NDV-based antineoplastic therapy has demonstrated to elicit a robust long-term anti-tumor immunity in different preclinical tumor models [38,39]. In the present work, we present a new glioma stratification system based on the *type I-IFN* cluster genomic expression, that could classify patients as potential responders and non-responders to NDV therapy and possibly other oncolytic viruses.

As more than 50% of GBM patients present a loss of heterozygosity (LOH) at chromosome 9p21 which harbors *IFN-I* cluster [40,41], the six CSCs models used in this study are representative of the equivalent human phenotype.

The results of the study indicate that there are two clusters of GBM CSCs with different degrees of susceptibility to NDV infection. One in which the NDV replication process occurs completely and another cluster in which the replication is abrogate. These differences could be explained by the induction of *type I IFNs*. The secretion of these *type I IFN* cytokines can trigger an antiviral immune response which activates *type I IFN* signaling and the induction of ISGs that blocks viral replication [42]. As it is known that the *IFN-I* signaling regulates the expression of *TNF-α, FAS* ligand and *TRAIL* [43], that effect is related with the high levels of those pro-apoptotic genes expressed by the cluster composed by GBM18, GBM38, and GBM123 upon NDV infection. Moreover, GBM tumors with high levels of *IFN-β* are more sensitive to TMZ and have better overall survival [44]. Thus, for the poor prognosis group composed by the *type I IFN* non-competent or deficient CSCs, NDV therapy could serve as a potential therapeutic option.

In line with previous researchers that demonstrate the NDV oncolytic effect in orthotopic GL261 mouse glioma model with a meaningful impact on overall survival [15], we observed a clear effect in tumor size reduction in *type I IFN*-deficient cell line (GBM27) model as compared to the *type I IFN*-competent tumors. Unfortunately, high toxicity levels were developed in these immunocompromised nude mice and we were unable to monitor survival rates. To demonstrate that the toxicity detected was due to the lack of cell-mediated immunity in that mice strain, we treated CD1 mice with the same virus concentration and no harm was detected (Appendix A). However, additional research is required in immunocompetent murine models to further confirm these data.

The use of NDV oncolytic virotherapy in several clinical trials has been presented as a safety strategy for treating solid tumors [45], such as GBM [46,47], melanoma [48], renal cell carcinoma [49], and ovarian cancer [50], among others. This therapy seems to have a high response heterogeneity in patients. Based on the data presented here, prior characterization of the type of tumor with regard to the presence or absence of the *type I IFN* cluster prior to the treatment with NDV (and possibly with other oncolytic tumors) is strongly suggested in order to better predict the patients’ response. 

Other oncolytic viruses have shown efficacy in treating GBM and pediatric brain tumors mediated by direct oncolysis followed by an immune response [51,52,53]. Notably, the use of virotherapy together with radio and chemotherapy has produced a synergistic effect [54].

Tumors that lack *type I IFN* induction response will be more receptive to replicate the virus and get a better oncolytic activity as compared to those that are *type I IFN*-competent. In the case of the *type I IFN*-competent tumors, the viral replication would be limited, and the oncolytic effect less effective. Still, the induction of the immune response against the infected cells may trigger the activation of immune cells against the tumor and show oncolytic effect.

*Type I IFN* induction has important roles in tumor immunogenicity such as antigen presentation, maturation of dendritic cells (DCs), activation and survival of memory cytotoxic T lymphocytes, NK cell activation, or neutrophil recruitment [55,56]. Loss of *type I IFN* cluster by cancer cells partially disables the ability of viral infection to trigger tumor interaction with the adaptive immune response [57]. In this scenario, production of *type I IFN* relies on non-tumor cells such as tumor infiltrated macrophages and DC or other cells present at the tumor microenvironment that do not have the genetic alterations of the cancer cells. 

The tumor microenvironment plays a key role in participating in the response and secretion of *type I IFN* to the tumor stroma [58]. As we have seen, there are GBM CSCs with *CDKN2A-Type I IFN* gene cluster codeletion, that could activate the signaling pathway in response to exogenous *IFN-β* and could be potentially activated by other *type I IFNs*. In the *type I IFN*-deficient tumors, NDV infection could trigger *type I IFN* production by other tumor-infiltrating immune cells. *IFN* production in this context will induce *type I IFN* signaling that can trigger ISG expression and induce a program in the tumor microenvironment that could control the virus replication and also generate an antitumor immune response. As a consequence, *IFN* produces anti-proliferative effects, described since 1962 [59], and decreases stem properties as it has been reported in other pathologies [60]. These antiproliferative effects are not the same in all our GBM CSCs, which opens another factor of heterogenicity in the oncolytic response to the virus. As mentioned above, NDV therapy reduced CSCs self-renewing capacity and increased their ability to differentiate. Targeting these mechanisms could be used in differentiation-based cancer therapy, triggering synergistic antitumor effects [61].

We have confirmed that *IFN I* responses underlie GBM CSCs selective replication of NDV. Based on this GBM classification, it is possible to employ alternative strategies that could bypass the *IFN*-competent GBMs. We have proved that *type I IFN*-competent GBM cells refractory to NDV can become susceptible to the virus if the *type I IFN* system is inhibited in those cells by other means. In this case, we have used a recombinant NDV expressing the influenza *NS1* gene that blocks *IFN* induction (rNDV-NS1) [62]. This recombinant virus has an oncolytic effect in both *type I IFN*-competent or deficient CSCs. rNDV-NS1 has been used before, showing improved oncolytic efficacy as a therapeutic and safe agent as compared to a NDV control [24,63]. Increasing the oncolytic potential of NDV by inserting *type I IFN* antagonistic proteins will need further characterization in GBM mouse models as the immune privileged brain environment response to these modified viruses is uncertain. 

In conclusion, our data highlight the importance of a glioma patient stratification based on the *CDKN2A-IFN* gene cluster deletion. That new classification could predict the patients that might potentially respond to NDV oncolytic therapy and possibly use it as a sensitivity biomarker to other oncolytic viruses [64]. For optimal patient management in GBM clinical practice, accurate *CDKN2A* CNA analysis prior to treatment is recommended to identify potential NDV non-responder patients and exclude them. The combination of conventional therapies with NDV oncolytic agent may provide a new promising approach for reducing GBM recurrence and improving the overall survival rate.

## Figures and Tables

**Figure 1 cells-09-01405-f001:**
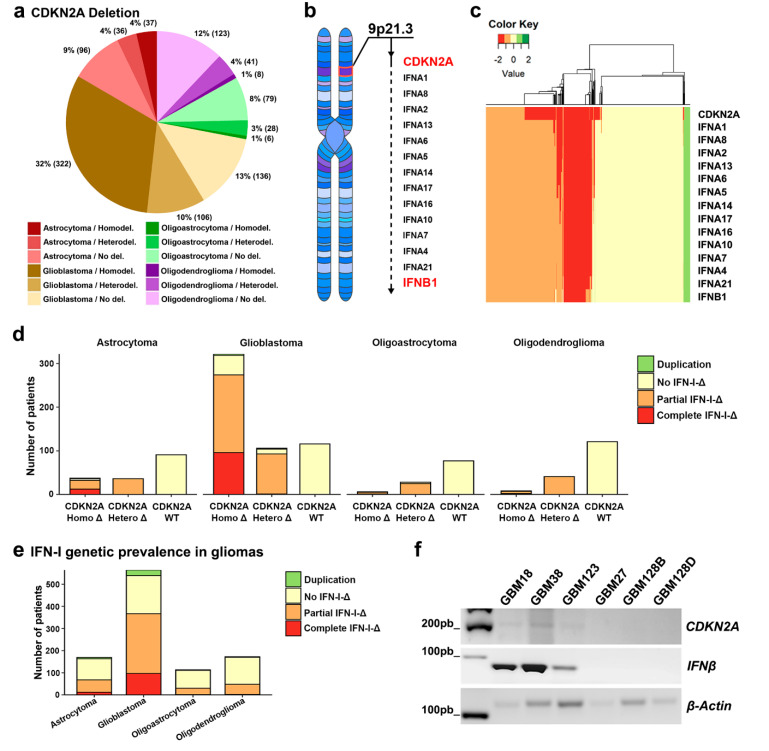
*CDKN2A* and *IFN* gene cluster prevalence in glioma tumors. (**a**) CDKN2A copy number analysis in glioma samples (total samples: 1018; astrocytoma *n* = 169, oligoastrocytoma *n* = 113, oligodendroglioma *n* = 172, and glioblastoma (GBM) *n* = 564). (**b**) Location of the *CDKN2A* and *IFN-I* cluster genes at the 9p21 chromosomal region. (**c**) Clustering of gliomas based on the amplification/deletion state of *CDKN2A* and *type I IFN* gene. (**d**) *CDKN2A* and *IFN-I* genomic copy number variation analysis in multiple glioma patients (**e**) *Type I IFN* aberration incidence in glioma patients. (**f**) *CDKN2A*, *IFNβ*, and *β-Actin* sequence amplification from GBM CSCs’ genomic DNA.

**Figure 2 cells-09-01405-f002:**
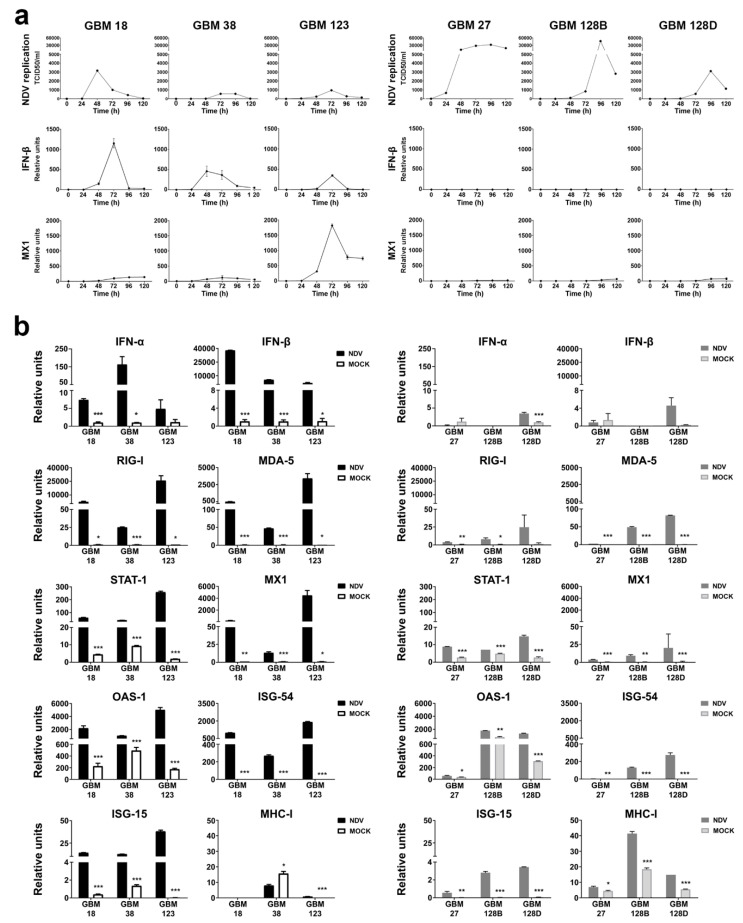
Newcastle disease virus (NDV) replication kinetics and antiviral response analysis in type I-competent (GBM18, GBM38, and GBM123) and non-competent CSCs (GBM27, GBM128, and GBM128D). (**a**) NDV growth curve at low multiplicity of infection (MOI) (0.01) over 120 h. *IFN-β* and MX1 qRT-PCR expression during viral infection. (**b**) *qRT-PCR* gene expression of *IFN-α*, *IFN-β*, RIG-I, MDA5, STAT1, MX1, OAS, ISG-54, ISG-15, and MHC-I. CSCs were infected at a MOI of 4 and RNA was extracted after 24 h. Relative units (mRNA copy number)are the mean ± SD. * *p* < 0.05, ** *p* < 0.01, *** *p* < 0.001.

**Figure 3 cells-09-01405-f003:**
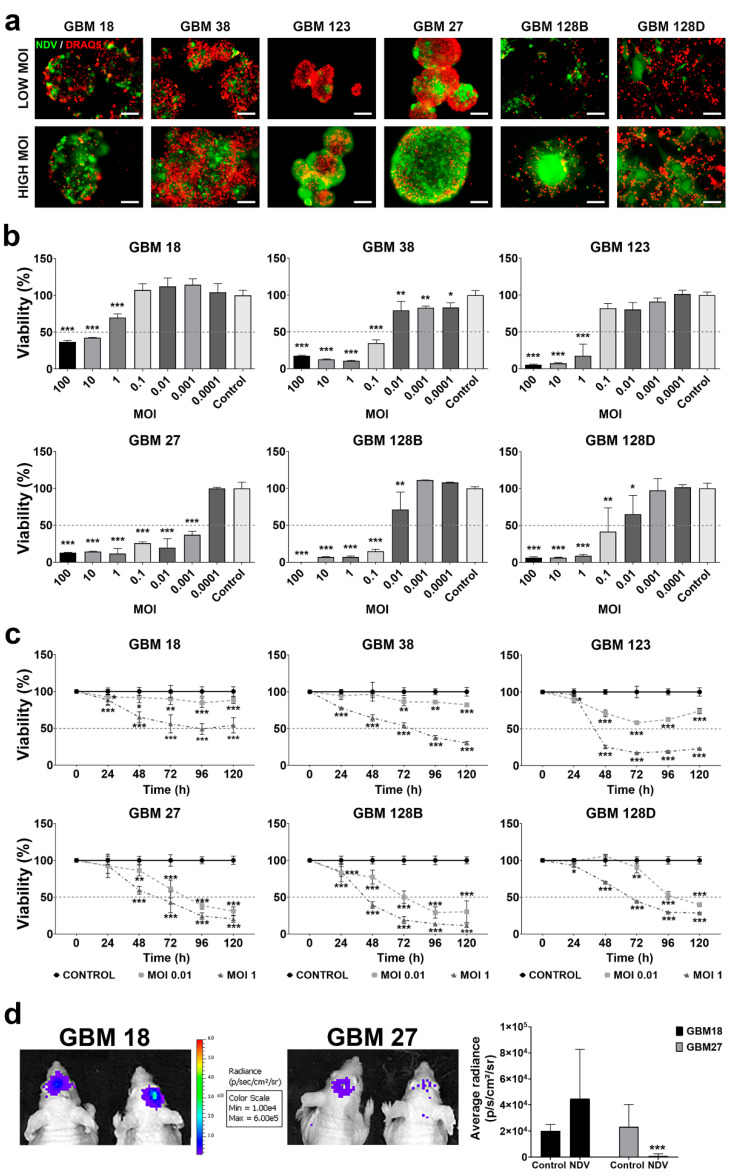
Infectivity of NDV in GBM CSCs. (**a**) Representative DRAQ5 staining of DNA content and rNDV-GFP fluorescence image. Scale bar = 100 µm. (**b**) CSCs NDV infection at different MOIs (100, 10, 1, 0.1, 0.01, 0.001, 0.0001, and 0). Percent of viability was measured by MTS assay at 144 h (**c**) CSCs were infected at a MOI of 0.01 or 1 and cell viability was analyzed every 24 h over a time course of 120 h. (**d**) Intracranial tumor burden was assessed by IVIS imaging system at day 7 post NDV injection. Mean area was calculated by Image J. Results * *p* < 0.05, ** *p* < 0.01, *** *p* < 0.001.

**Figure 4 cells-09-01405-f004:**
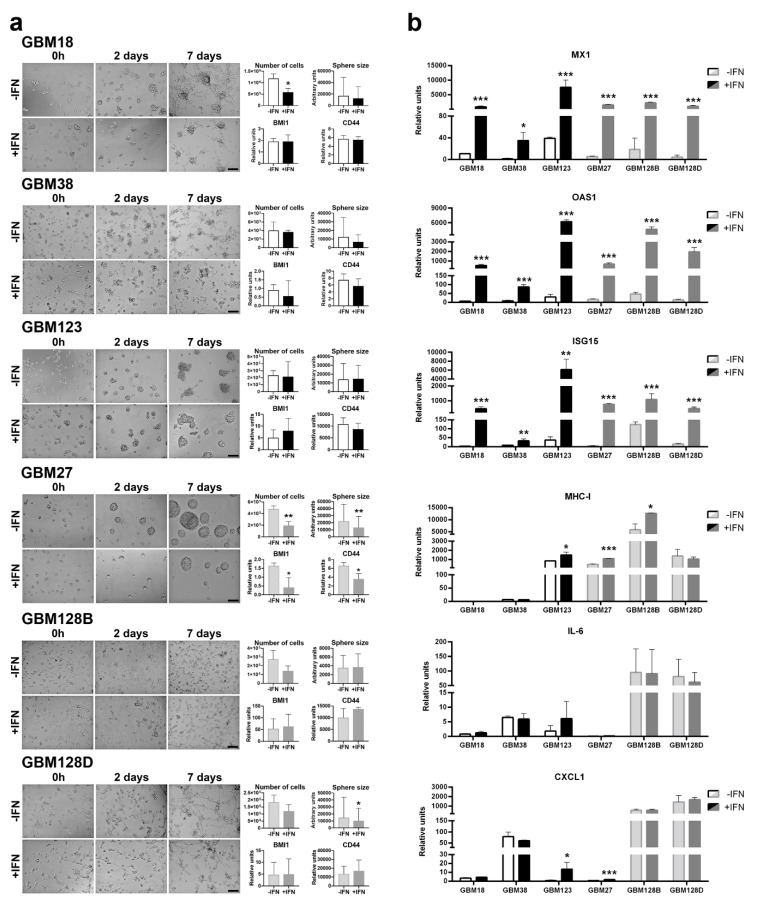
*IFN-β* treatment. (**a**) CSCs images were taken at 0, 2, and 7 days after exogenous *IFN-β* treatment. Scale bar: 200 µm. Trypan blue was used to check cell viability and sphere diameter was measured with CellProfiler software at 7 days. RNA was isolated after 24 h of treatment, and BMI1 and CD44 were analyzed. (**b**) mRNA relative units were measured. Graphics are present as the mean ± SD. * *p* < 0.05, ** *p* < 0.01, *** *p* < 0.001.

**Figure 5 cells-09-01405-f005:**
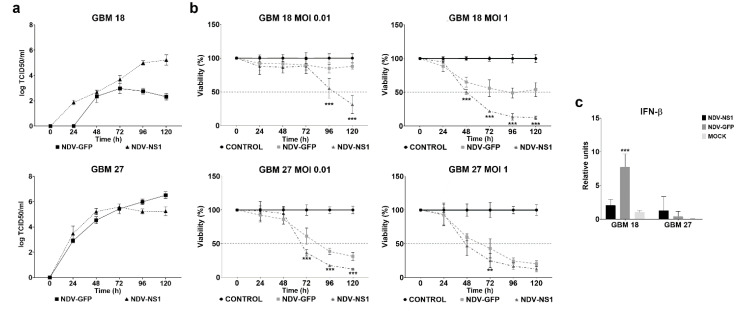
NDV-NS1 restored oncolytic activity in competent CSCs model. (**a**) rNDV-GFP and rNDV-NS1 viral growth curves were monitored for 120 h in GBM18 and GBM27 (**b**) GBM18 and GBM27 were exposed to NDV-GFP and NDV-NS1 MOI 1 and 0.01. Cell viability was measured every 24 h during 120 h (**c**) 100,000 CSCs per well were seeded in a 12 well plate and infected with NDV-GFP and NDV-NSI MOI 0.01. mRNA was isolated at 72 h. Error bar represents standard deviation. ** *p* < 0.01, *** *p* < 0.001.

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
