# Peer review of "Newcastle Disease Virus (NDV) Oncolytic Activity in Human Glioma Tumors Is Dependent on CDKN2A-Type I IFN Gene Cluster Codeletion"

_cells, 2020, doi:10.3390/cells9061405_

Round 1
Reviewer 1 Report
The Authors present a study of human GBM CSC’s susceptibility to oncolytic NDV. The manuscript is generally clearly written and only minor language revision is needed. The use of human glioma samples makes the results interesting and relevant to the oncolytic virotherapy field.
Questions/comments that should be addressed by the Authors:
- The Authors should provide explanation to use of DRAQ5 probe was used in infectivity assay.
- Cell viability measurements with MTS assay should be complemented with more direct detection of cell death. E.g. by measuring caspase activity.
- Line 184: “Then we focused in glioma IDH-WT for analysis.” Does this mean that the data presented in Figure 1 is form IDH-WT gliomas? If so, are the results similar in IDH-mutated gliomas? Please clarify?
- The meaning/relevance of Figure 1c is not clear to me. Please provide more detailed explanation in the results part.
- Figure 1f results should be quantified and normalized to b-actin.
- GBM128D is claimed by the Authors to be “type-I-non competent” but according to Figure 1b is producing low IFN-a and IFN-b when infected with NDV. Please clarify this.
- NDV seems to replicate poorly in GBM128D (figure 2a) despite of low IFN-response. Have the authors tested for any alternative explanations for the NDV susceptibility? For example, have the authors checked the expression of NDV receptor on the analysed GBM samples?
- Related to comment 8, (as also discussed by the Authors) type-I-IFN in the tumor microenvironment can be also produced by tumor-infiltrating immune cells or other stromal cells. This can interfere with NDV replication in IFN-I responsive GBM cells even if their own capability to produce IFN-I is low. The authors should therefore also investigate the replication of NDV in IFN-I exposed cells. Is there a possible correlation between ISG expression level and NDV replication?
- Cell images in Figure 4 are too small. Larger high-resolution images should be provided.
- Construction of rNDV-NS1 virus must be explained. Schematic presentation of the virus genome should be provided.
- Addition of IFN-I antagonising NS1 gene into the virus raises safety concern as it likely reduces cancer cell selectivity. The Author’s should discuss the potential increase in biosafety level and possible strategies to improve safety.
- Figure 5b both GBM 18 panels are labelled “GBM 18 MOI 0.01”. Please, indicate which panel shows result of MOI 1 infection.
- Line 270: “CSCs presented high resistance to die when they were infected at a low MOI.” Resistance to die in this case should be changed to resistance to virus.
- Line 271: “loss on viability in… was remarkably higher” should be rewritten to make it easier to read. Please simplify.
- Line 271: Importantly, GBM27 272 presented a delayed susceptibility to the virus that is significative only after 72 hours of NDV 273 infection”. It is not clear what the Authors mean by this and why this is important. Please clarify.
- Production of luciferase expressing CSCs must be provided. In addition, IFN-I competence of these cell must be analysed to make sure the phenotype is not affected.
- Luminescence signal from mice (Figure 3d) should be quantified using calibrated physical units (p/sec/cm2/sr) instead of area.
- The authors should show viral replication in the tumors in vivo (Figure 3d).
- Line 373: “ We and others have demonstrated that more than 50% of GBM patients present a Loss of Heterozygosity (LOH) at chromosome 9p21 which harbors IFN-I cluster [38,39]. For this reason, we believe that our CSCs models are reliable, and representative of the population analyzed.” It is not completely clear what the Authors mean by this statement. Does this refer to selection of IFN-competent and non-competent CSCs for the experiments? Please clarify.
- Line 435: “For optimal patient management in clinical practice, accurate CDKN2A CNA analysis prior to treatment is recommended and GBM patients could be reliably divided into NDV responder and non-responder groups.” This is an overstatement, as this kind of analysis does include responsiveness to exogenous IFN-I which could still limit NDV replication in the glioma cells.
- The Authors should provide more detailed explanation on how they envision the patient DNA analysis to be performed in real practise. Would it require multiple samples to account possible intratumor heterogeneity etc. How feasible would it be to include this analysis into the standard patient care in order to select possible candidates for NDV therapy?
- Related to comment 21, Therapeutic success of oncolytic virotherapy is likely strongly linked to induction of immunogenic cancer cell death which in turn is linked to expression of DAMPs and inflammatory cytokines (such as IFN-a and IFN-b) by the infected cancer cells. The Authors should speculate how the loss of IFN-I expression in the GBM cells might affect the immunotherapeutic potency of NDV (or OVs in general).
- Line 437: “The combination of conventional therapies with NDV oncolytic agent may provide a new promising approach for reducing GBM recurrence and improving the overall survival rate.” Do the authors have any preclinical results to support this?
Author Response
Reviewer #1:
1) The Authors should provide explanation to use of DRAQ5 probe was used in infectivity assay.
We appreciate the concern of the Reviewer #1, as he/she knows DRAQ5™ is a cell permeable DNA dye that can be used in non-fixed live cells in combination with common labels such as GFP or FITC. In the infectivity assay (Figure 3a) we used this dye directly to avoid artifacts during cell fixation.
To clarify that, we have added a new sentence in “Materials and Methods” section, line 125-126.
2) Cell viability measurements with MTS assay should be complemented with more direct detection of cell death. E.g. by measuring caspase activity.
We agree with the Reviewer that adding another technique to measure cell death would increase the manuscript impact. For this reason, we have evaluated the apoptosis rate using annexin V that binds to phosphatidylserine and a DNA dye (7-AAD).
- Annexin V +, 7-AAD – were considered early apoptotic cells with an intact cell membrane.
- Annexin V +, 7-AAD+ were considered dead cells.
Apoptosis rate results at 120h after NDV infection are shown in Figure S4.
References:
Adan, A.; Alizada, G.; Kiraz, Y.; Baran, Y.; Nalbant, A. Flow cytometry: basic principles and applications. Crit. Rev. Biotechnol. 2017, 37, 163–176, doi:10.3109/07388551.2015.1128876.
3) Line 184: “Then we focused in glioma IDH-WT for analysis.” Does this mean that the data presented in Figure 1 is form IDH-WT gliomas? If so, are the results similar in IDH-mutated gliomas? Please clarify?
We thank the Reviewer #1 for raising an important question in the field of gliomas. The last World Health Organization Classification of Tumors of the Central Nervous System (2016) established the use of both phenotypic and genotypic features to classify CNS tumors. Diffuse gliomas are now divided into three categories based in IDH status: IDH-mutant, IDH-wildtype or NOS (Not Otherwise Specificied), if there is not enough information to assign IDH code. Following the Reviewer advice, we have included IDH-mutant patients in the analysis showing similar results as with the wildtype cohort. We have made the necessary arrangements in the main text, in figure 1 and its legend.
References:
Louis, D.N.; Perry, A.; Reifenberger, G.; von Deimling, A.; Figarella-Branger, D.; Cavenee, W.K.; Ohgaki, H.; Wiestler, O.D.; Kleihues, P.; Ellison, D.W. The 2016 World Health Organization Classification of Tumors of the Central Nervous System: a summary. Acta Neuropathol. 2016, 131, 803–820, doi:10.1007/s00401-016-1545-1.
4) The meaning/relevance of Figure 1c is not clear to me. Please provide more detailed explanation in the results part.
We agree with the Reviewer#1 that more explanation of the Figure 1c could help to understand the manuscript. The aim of this gene clustering is to identify if CDKN2A and type I IFN genes are behaving coordinate. After the analysis we noticed that CDKN2A profiles were strongly correlated with type I IFN genes. Most of the patients grouped together in different clusters. The color is used to represent changes in gene pattern. In this heatmap, green represents gene amplification, red gene deletion and yellow the wild type profile.
Following his/her advice, we have added the following sentence in the Results section line 212 page 6: “Gene clustering pointed out that CDKN2A gene state is coordinate with type I IFN genes”.
5) Figure 1f results should be quantified and normalized to b-actin.
Although we understand the concern of Reviewer #1, we would like to drive the attention to the Figure 1f legend “CDKN2A, IFNβ and β-Actin sequence amplification from GBM CSCs’ genomic DNA”. The aim of this conventional PCR was to study the presence or absence of these genes at the genomic level (gDNA). If we had wanted to show the amount of expression, we would have used a quantitative real-time PCR that is widely established for assessing cDNA expression normalizing with a housekeeping gene. In any case, we have quantified gel bands by Image Lab software v 6.0.1 as shown below and added the information to Supplementary section (Figure S2). However, if the Reviewer#1 still considers that this quantification should be included in the main text, we will be willing to follow his directions.
Figure S2. gDNA quantification of CDKNA and IFN-β. β-actin was used as reference control.
References:
Yun, J.J.; Heisler, L.E.; Hwang, I.I.L.; Wilkins, O.; Lau, S.K.; Hyrcza, M.; Jayabalasingham, B.; Jin, J.; McLaurin, J.A.; Tsao, M.S.; Der, S.D. Genomic DNA functions as a universal external standard in quantitative real-time PCR. Nucleic Acids Res. 2006, 34, 1–10, doi:10.1093/nar/gkl400.
6) GBM128D is claimed by the Authors to be “type-I-non competent” but according to Figure 1b is producing low IFN-a and IFN-b when infected with NDV. Please clarify this.
We guess that instead of Figure 1b, the Reviewer#1 refers to Figure 2b. In that case, we wanted to force the viral effect in the host CSCs gene expression, so we infected them with a higher multiplicity of infection (MOI) of 4. Nonetheless, only a 4-fold increase was observed in GBM128D IFN-β expression after NDV infection, this value could be considered as insignificant if we compared it with the values reached in type-I competent CSC group (3.7 x 104 times).
7) NDV seems to replicate poorly in GBM128D (figure 2a) despite of low IFN-response. Have the authors tested for any alternative explanations for the NDV susceptibility? For example, have the authors checked the expression of NDV receptor on the analysed GBM samples?
As the Reviewer#1 knows the entry of most of paramyxoviruses in the cells is not mediated by a specific receptor, it occurs by direct fusion of the viral envelope with the plasma membrane. In the case of NDV, the glycoprotein hemagglutinin-neuraminidase (HN) recognizes sialic-conjugates at the cell surface. Despite the lack of a specific receptor, there are some sensors (RIG-I-like receptors) that detect viral RNA and activate the response signaling, for this reason we have analyzed its expression under NDV infection in Figure 2b.
References:
Sánchez-Felipe, L.; Villar, E.; Muñoz-Barroso, I. Entry of Newcastle Disease Virus into the host cell: Role of acidic pH and endocytosis. Biochim. Biophys. Acta - Biomembr. 2014, 1838, 300–309, doi:10.1016/j.bbamem.2013.08.008.
Nistal-Villán, E.; Rodríguez-Garciá, E.; Di Scala, M.; Ferrero-Laborda, R.; Olaguë, C.; Vales, Á.; Carte-Abad, B.; Crespo, I.; Garciá-Sastre, A.; Prieto, J.; Larrea, E.; González-Aseguinolaza, G. A RIG-I 2CARD-MAVS200 Chimeric Protein Reconstitutes IFN-β Induction and Antiviral Response in Models Deficient in Type i IFN Response. J. Innate Immun. 2015, 7, 466–481, doi:10.1159/000375262.
8) Related to comment 8, (as also discussed by the Authors) type-I-IFN in the tumor microenvironment can be also produced by tumor-infiltrating immune cells or other stromal cells. This can interfere with NDV replication in IFN-I responsive GBM cells even if their own capability to produce IFN-I is low. The authors should therefore also investigate the replication of NDV in IFN-I exposed cells. Is there a possible correlation between ISG expression level and NDV replication?
We really thank the Reviewer#1 for providing such useful suggestion. We agree that evaluating NDV replication after IFN treatment would certainly be of interest. Indeed, we and others have demonstrated that NDV replication is depending of type I IFN signaling pathway and is well documented in the literature that the expression of ISGs affects viral entry into cells. For this reason, we have measured in Figure 4b the induction of some ISGs as the oligoadenylate synthetase (OAS) which main function is to detect foreign RNA or the MX1 and ISG15. Our data demonstrate that both groups of CSCs are able to induce antiviral response. Based on these data, we can conclude that some of the CSC whose IFN cannot be induced by NDV, can be stimulated when treated with IFN-β.
References:
Ginting, T.E.; Christian, S.; Larasati, Y.O.; Suryatenggara, J.; Suriapranata, I.M.; Mathew, G. Antiviral interferons induced by Newcastle disease virus (NDV) drive a tumor-selective apoptosis. Sci. Rep. 2019, 9, 1–10, doi:10.1038/s41598-019-51465-6.
Malterer, M.B.; Glass, S.J.; Newman, J.P. Interferon-stimulated genes: A complex web of host defenses. Annu. Rev. Immunol. 2014, 44, 735–745, doi:10.1038/jid.2014.371.
9) Cell images in Figure 4 are too small. Larger high-resolution images should be provided.
We really appreciate and thank this observation from Reviewer #1, we have increased cell images size in Figure 4, however if he/she considers that they still too small we could divide the Figure 4 into two different images as seen below:
1
2
10) Construction of rNDV-NS1 virus must be explained. Schematic presentation of the virus genome should be provided.
We sincerely appreciate Reviewer #1 for his/her insightful suggestion. Following the advice, and to facilitate manuscript comprehension, we have added a new schematic figure in the Supplementary section (Figure S1).
Figure S1: Schematic representation of rNDV viruses. (a) wt-NDV genome contains six genes that codifies for nucleocapside (N), phospho (P), matrix (M), fusion (F), hemagglutinin-neuraminidase (HN) and large (l) proteins. (b) Green fluorescent protein (GFP) or (c) Influenza virus NS1 gene were inserted between P and M genes.
11) Addition of IFN-I antagonising NS1 gene into the virus raises safety concern as it likely reduces cancer cell selectivity. The Author’s should discuss the potential increase in biosafety level and possible strategies to improve safety.
We and others have previously demonstrated that reverse genetics systems are safety to manipulate the NDV genome. Although, it has been published that some insertion could lose the avian host restriction, recombinant NDV (rNDVs) contains foreign genes as NS1 that increases NDV replication rate in human cells. We have specifically used for the Figure 5 rNDV viruses that have not been mutated at the furin cleavage site, retaining the original lentogenic characteristics of NDV LaSota. This recombinant virus is safer that the F3aa versions.
References:
Park, M.-S.; Garcia-Sastre, A.; Cros, J.F.; Basler, C.F.; Palese, P. Newcastle Disease Virus V Protein Is a Determinant of Host Range Restriction. J. Virol. 2003, 77, 9522–9532, doi:10.1128/jvi.77.17.9522-9532.2003.
Zamarin, D.; Martínez-Sobrido, L.; Kelly, K.; Mansour, M.; Sheng, G.; VigilA, A.; García-Sastre, A.; Palese, P.; Fong, Y. Enhancement of oncolytic properties of recombinant newcastle disease virus through antagonism of cellular innate immune responses. Mol. Ther. 2009, 17, 697–706, doi:10.1038/mt.2008.286.
12) Figure 5b both GBM 18 panels are labelled “GBM 18 MOI 0.01”. Please, indicate which panel shows result of MOI 1 infection.
We have labelled it correctly.
13) Line 270: “CSCs presented high resistance to die when they were infected at a low MOI.” Resistance to die in this case should be changed to resistance to virus.
We completely agree with Reviewer #1 and thank him/her for pointing out this typo.
14) Line 271: “loss on viability in… was remarkably higher” should be rewritten to make it easier to read. Please simplify.
Following his/her suggestion we have re-written the sentence, which now is:
“In contrast, GBM27, GBM128B and GBM128D non-competent cells were more susceptible to the virus infection at a MOI of 0.01 than the IFN competent cells over the extent of the study”.
15) Line 271: Importantly, GBM27 presented a delayed susceptibility to the virus that is significative only after 72 hours of NDV infection”. It is not clear what the Authors mean by this and why this is important. Please clarify.
We understand the concern of the Reviewer #1, as this sentence is quite confusing, we have modified and added:
“This effect was observed after 24h of NDV infection in GBM128B and GBM128D. Instead, GBM27 presented a delayed susceptibility to the virus that is significative only after 48 hours of NDV infection”.
16) Production of luciferase expressing CSCs must be provided. In addition, IFN-I competence of these cell must be analysed to make sure the phenotype is not affected.
We really appreciate the suggestion of the Reviewer#1, following his/her advice we have detailed it in Materials and Methods section 2.2. As GBM18 and GBM27 CSCs are widely characterized by our group we found that expression of the luciferase has no effect on either the in vitro proliferation, or on their ability to form tumors after the orthotopic transplantation.
References:
García-Romero, N.; González-Tejedo, C.; Carrión-Navarro, J.; Esteban-Rubio, S.; Rackov, G.; Rodríguez-Fanjul, V.; Oliver-De La Cruz, J.; Prat-Acín, R.; Peris-Celda, M.; Blesa, D.; Ramírez-Jiménez, L.; Sánchez-Gómez, P.; Perona, R.; Escobedo-Lucea, C.; Belda-Iniesta, C.; Ayuso-Sacido, A. Cancer stem cells from human glioblastoma resemble but do not mimic original tumors after in vitro passaging in serum-free media. Oncotarget 2016, 7, doi:10.18632/oncotarget.11676.
García-romero, N.; Carrión-navarro, J.; Esteban-Rubio, S.; Lazaro-Ibanez, E.; Peris-Celda, M.; Alonso, M.M.; Guzman-De-Villoria, J.; Fernández-Carballal, C.; de Mendivil, A.O.; García-Duque, S.; Escobedo-Lucea, C.; Prat-Acin, R.; Belda-Iniesta, C.; Ayuso-Sacido, A.; Mendivil, A.O. De; García-Duque, S. DNA sequences within glioma-derived extracellular vesicles can cross the intact Blood-Brain Barrier and be detected in peripheral blood of patients. Oncotarget 2016, 8, 1–13, doi:10.18632/oncotarget.13635.
17) Luminescence signal from mice (Figure 3d) should be quantified using calibrated physical units (p/sec/cm2/sr) instead of area.
We agree with the Reviewer#1 that area is not the best option to quantify the images obtained by the IVIS luminometer system. As explained in Results section, “when the tumors reached an average radiance of 6.77 x 103 [p/s/cm²/sr], they were injected with 5 x 104 UI/ ml of rNDV-GFP”, so we have modified the units to evaluate tumor burden in Figure 3d.
18) The authors should show viral replication in the tumors in vivo (Figure 3d).
We perfectly understand the concern from Referee #1. As the reviewer knows is quite difficult to follow the fluorescence of the GFP-NDV in vivo several days after infection. However, we tried to evaluate it and to observed NDV replication in the tissue sections. As seen below we detected a slight signal non-specific, that we believe that could be related to the specificity of antibody labeling or due to an inadequate fixation.
19) Line 373: “ We and others have demonstrated that more than 50% of GBM patients present a Loss of Heterozygosity (LOH) at chromosome 9p21 which harbors IFN-I cluster [38,39]. For this reason, we believe that our CSCs models are reliable, and representative of the population analyzed.” It is not completely clear what the Authors mean by this statement. Does this refer to selection of IFN-competent and non-competent CSCs for the experiments? Please clarify.
We thank the Reviewer #1 for highlighting this paragraph which was quite confusing. The main idea we wanted to transmit is that the six CSCs models used in our study represent the relative frequency of IFN-competent and non-competent phenotypes observed in GBM patients.
We have checked the manuscript and clarified that “As more than 50% of GBM patients present a Loss of Heterozygosity (LOH) at chromosome 9p21 which harbors IFN-I cluster [38,39], the six CSCs models used in this study are representative of the equivalent human phenotype”.
20) Line 435: “For optimal patient management in clinical practice, accurate CDKN2A CNA analysis prior to treatment is recommended and GBM patients could be reliably divided into NDV responder and non-responder groups.” This is an overstatement, as this kind of analysis does include responsiveness to exogenous IFN-I which could still limit NDV replication in the glioma cells.
We have softened the sentence that now says: “For optimal patient management in GBM clinical practice, accurate CDKN2A CNA analysis prior to treatment is recommended to identify potential NDV non-responder patients and exclude them”.
21) The Authors should provide more detailed explanation on how they envision the patient DNA analysis to be performed in real practise. Would it require multiple samples to account possible intratumor heterogeneity etc. How feasible would it be to include this analysis into the standard patient care in order to select possible candidates for NDV therapy?
As we have explained before, since 2016 molecular markers are used in clinical practice. In fact, as The Cancer Genome Atlas (TCGA) associates the classical GBM subtype with CDKN2A loss, the DNA copy number analysis is now routinely performed in most of the institutions only to classify the tumor sample. So, this information could be used to select potential responders to NDV treatment.
References:
Louis, D.N.; Perry, A.; Reifenberger, G.; von Deimling, A.; Figarella-Branger, D.; Cavenee, W.K.; Ohgaki, H.; Wiestler, O.D.; Kleihues, P.; Ellison, D.W. The 2016 World Health Organization Classification of Tumors of the Central Nervous System: a summary. Acta Neuropathol. 2016, 131, 803–820, doi:10.1007/s00401-016-1545-1.
Verhaak, R.G.W.; Hoadley, K.A.; Purdom, E.; Wang, V.; Qi, Y.; Wilkerson, M.D.; Miller, C.R.; Ding, L.; Golub, T.; Mesirov, J.P.; Alexe, G.; Lawrence, M.; O’Kelly, M.; Tamayo, P.; Weir, B.A.; Gabriel, S.; Winckler, W.; Gupta, S.; Jakkula, L.; Feiler, H.S.; Hodgson, J.G.; James, C.D.; Sarkaria, J.N.; Brennan, C.; Kahn, A.; Spellman, P.T.; Wilson, R.K.; Speed, T.P.; Gray, J.W.; Meyerson, M.; Getz, G.; Perou, C.M.; Hayes, D.N. Integrated Genomic Analysis Identifies Clinically Relevant Subtypes of Glioblastoma Characterized by Abnormalities in PDGFRA, IDH1, EGFR, and NF1. Cancer Cell 2010, 17, 98–110, doi:10.1016/j.ccr.2009.12.020.
22) Related to comment 21, Therapeutic success of oncolytic virotherapy is likely strongly linked to induction of immunogenic cancer cell death which in turn is linked to expression of DAMPs and inflammatory cytokines (such as IFN-a and IFN-b) by the infected cancer cells. The Authors should speculate how the loss of IFN-I expression in the GBM cells might affect the immunotherapeutic potency of NDV (or OVs in general).
We completely agree with the Reviewer statement that the cell response to a viral infection is the production of Type I IFN.
The loss of IFN-I expression by the infected cell can have important consequences in the immunogenicity and immune stimulatory effects of the tumor such as the maturation of DCs, activation and survival of memory cytotoxic T lymphocytes, NK cell activation, or neutrophil recruitment. Inability to induce IFN-I upon infection by OV, reduces the chances to have a full activation of the abovementioned cells.
We have introduced this concept in the discussion section.
References:
Zitvogel L1, Galluzzi L2, Kepp O3, Smyth MJ4, Kroemer G5. Type I interferons in anticancer immunity. Nat Rev Immunol. 2015 Jul;15(7):405-14. PMID: 26027717
Parker BS, Rautela J, Hertzog PJ. Antitumour actions of interferons: implications for cancer therapy. Nat Rev Cancer. 2016 Mar;16(3):131-44. PMID: 26911188
Medrano RFV, Hunger A, Mendonça SA, Barbuto JAM, Strauss BE. Immunomodulatory and antitumor effects of type I interferons and their application in cancer therapy.Oncotarget. 2017 Jul 25;8(41):71249-71284.
Russell, Stephen J et al. “Oncolytic virotherapy.” Nature biotechnology vol. 30,7 658-70. 10 Jul. 2012, doi:10.1038/nbt.2287.
Singh PK, Doley J, Kumar GR, Sahoo AP, Tiwari AK. Oncolytic viruses & their specific targeting to tumour cells. Indian J Med Res. 2012 Oct;136(4):571-84.
23) Line 437: “The combination of conventional therapies with NDV oncolytic agent may provide a new promising approach for reducing GBM recurrence and improving the overall survival rate.” Do the authors have any preclinical results to support this?
We thank the Reviewer #1 for raising an important question in the field of GBM treatment. As he/she knows, the combination of NDV with other therapies has been well documented by other groups, showing that its use with temozolomide treatment has a synergic effect and increases overall survival in glioma mouse models. In GBM patients the combined NDV with dendritic-cell based therapy prolongs 7 months the OS. We have detailed in the introduction section.
References:
- Nesselhutet al., Improvement of dendritic cell therapy in glioblastoma multiforme WHO 4 by Newcastle disease virus. Journal of Clinical Oncology. 2011, 15, 2508-2508, doi: 10.1200/jco.2011.29.15_suppl.2508.
Bai, Y.; Chen, Y.; Hong, X.; Liu, X.; Su, X.; Li, S.; Dong, X.; Zhao, G.; Li, Y. Newcastle disease virus enhances the growth-inhibiting and proapoptotic effects of temozolomide on glioblastoma cells in vitro and in vivo. Sci. Rep. 2018,8, 1–12, doi:10.1038/s41598-018-29929-y.
Reviewer 2 Report
Summary: The oncolytic effect of Newcastle disease virus (NDV) was investigated in glioblastoma (GBM) to understand the interactions with type I interferon (IFN). In silico, ex vivo, and mouse models were investigated to quantify infection kinetics and the impact of IFN I loss on NDV replication competency in representative GBM cell lines.
1) Throughout, GBM27 is shown to have unique IFN gene signatures/expression and infection kinetics, but it was used as representative of non-competent CSCs. Why choose GBM27? Since there are only 6 cell samples, why not show results for all six over two representatives?
2) Is there any discernable difference in growth kinetics over a range of MOIs (rather than MOI=0.01, which was reported)?
3) Were alternate calculations of cell viability (i.e. live cells/(live cells + dead cells) assessed and why chose the survival rate formula on line 133?
4) There are a number of typos throughout:
mayor (line 253), susceptivility (line 263), higer (line 264), inefction (line 269), propose (line 344).
5) Fig.5 b, right panel, top should be GBM 18 MOI 1?
Author Response
Reviewer #2:
1) Throughout, GBM27 is shown to have unique IFN gene signatures/expression and infection kinetics, but it was used as representative of non-competent CSCs. Why choose GBM27? Since there are only 6 cell samples, why not show results for all six over two representatives?
We totally agree with the Reviewer#2 that using several GBM cell lines would increase the impact of the manuscript. As the reviewer knows, one of the most important issues in the process of designing an experiment is the cost effectiveness. For this reason, the first aim of our study was an in silico assay including more than 1000 patients. Once we focused our hypothesis, we analyzed 6 GBM CSCs and then decided to continue with one representative cell line of each group. As we have continued with in vitro, in vivo and patients’ studies, we believe that our study is well-founded. Moreover, the CSCs selected (GBM18 and GBM27) have been previously characterized by our group in the context of their original tumors and the in vitro and in vivo evolution.
However, if the Reviewer#2 still considers that we should add more cell lines, we will be willing to follow his directions.
References:
García-Romero, N.; González-Tejedo, C.; Carrión-Navarro, J.; Esteban-Rubio, S.; Rackov, G.; Rodríguez-Fanjul, V.; Oliver-De La Cruz, J.; Prat-Acín, R.; Peris-Celda, M.; Blesa, D.; Ramírez-Jiménez, L.; Sánchez-Gómez, P.; Perona, R.; Escobedo-Lucea, C.; Belda-Iniesta, C.; Ayuso-Sacido, A. Cancer stem cells from human glioblastoma resemble but do not mimic original tumors after in vitro passaging in serum-free media. Oncotarget 2016, 7, doi:10.18632/oncotarget.11676.
2) Is there any discernable difference in growth kinetics over a range of MOIs (rather than MOI=0.01, which was reported)?
We really appreciate and thank this observation from Reviewer #2. When we evaluated the % of viability at 144h using different MOIs (100, 10, 1, 0.1, 0.01, 0.001, 0.0001), we observed that in the IFN competent cells the cell viability using MOI of 0.1 was similar to the control (uninfected) cells. On the contrary in non-type I IFN competent CSCs decrease in cell viability is almost the same as with 1 MOI. These data are presented in Figure 3b.
3) Were alternate calculations of cell viability (i.e. live cells/(live cells + dead cells) assessed and why chose the survival rate formula on line 133?
We completely agree with Reviewer #2 in the sense that the word “survival rate” might not be the best option in this context. What we wanted to transmit is the percentage of cell viability relative to the mock-infected cells. In fact, the y-axes of the graphs were labelled correctly in Figure 3b-c and Figure 5b. We have corrected it in Material and Methods section 2.7.
4) There are a number of typos throughout:
mayor (line 253), susceptivility (line 263), higer (line 264), inefction (line 269), propose (line 344).
We are very grateful to Reviewer #2 for driving the attention on those typos, we have corrected and reviewed all the manuscript.
5) Fig.5 b, right panel, top should be GBM 18 MOI 1?
We really appreciate and thank the Reviewer #2 for pointing out this mistake, which is also a concern for Reviewer #1 and #3. We have labelled it correctly.
Reviewer 3 Report
Co-deletion at Ch9p21 (which results in loss of CDKN2A) and the type I Interferon (IFN) gene cluster are frequently observed in GBM. García-Romero et al., reasoned that the absence of the IFN genes could render GBM cells more susceptible to oncolytic viral therapies. Their hypothesis was tested in 6 primary GBM cell lines and in one immunodeficient mouse model. This work is of relevance to the field as it suggests that stratifying glioma patients based on CDKN2A and IFN codeletion could increase the therapeutic efficacy of oncolytic viral therapies.
One of the main issues is that the in vivo experiment, as currently presented, does not provide convincing evidence to support the authors theory. It is highly recommended that this experiment would be repeated with a lower MOI that would not lead to overall systemic toxicity but generate a significant extension of overall survival. This would certainly increase the significance of this work.
Also, there are numerous typos, grammatical errors and one of the figure panels is mislabeled (see below). This is overall distracting and at times, make it hard to evaluate the results (for ex. fig. 5B).
Additional comments:
- Introduction: Please provide more background on “type I IFN antagonistic protein such as the
non-structural 1 (NS1) protein of the influenza virus”.
- Methods: please provide more details about the media, supplements, source used to grow primary GBM cells
- When appropriate, please include proper statistical analysis in figure 1 (for ex. Fig. 1e).
- Figure 5b right panel is mislabeled, not clear which one shows an MOI of 1 since both are labeled MOI 0.01.
- There are numerous errors in the text, for ex:
Line 79 : “dead” should be cell death ?
Line 115: “GMB” should be GBM
Line 199 : “shown” should be shows
Lines 252, 253, etc.
Author Response
Reviewer #3:
1) One of the main issues is that the in vivo experiment, as currently presented, does not provide convincing evidence to support the authors theory. It is highly recommended that this experiment would be repeated with a lower MOI that would not lead to overall systemic toxicity but generate a significant extension of overall survival. This would certainly increase the significance of this work.
We agree with the Reviewer#3 that more in vivo assays could increase the impact of the manuscript. However, we have tried several infection units and observed the same toxicity, except when we injected CD1 mice (Figure S5). As we explained in the Discussion section, we believe that new experiments are required in other immuncompetent murine models to validate our results.
Figure S5. NDV toxicity assay. 10 Nude and 5 CD1 mice received a single intratecal dose of NDV. Animals were sacrified when their weight decreased more than 20% of their initial body weight.
Also, there are numerous typos, grammatical errors and one of the figure panels is mislabeled (see below). This is overall distracting and at times, make it hard to evaluate the results (for ex. fig. 5B).
We are very grateful to Reviewer #3 for driving the attention on the typos, we have corrected and reviewed all the manuscript.
Additional comments:
- Introduction: Please provide more background on “type I IFN antagonistic protein such as the non-structural 1 (NS1) protein of the influenza virus”.
We really appreciate this observation as more information about NS1 protein is required to facilitate manuscript comprehension. We have added it in the introduction section, and following Reviewer#1 recommendation we have made a schematic representation of rNDV viruses in Supplementary section (Figure S1).
References:
Zamarin, D.; Martínez-Sobrido, L.; Kelly, K.; Mansour, M.; Sheng, G.; VigilA, A.; García-Sastre, A.; Palese, P.; Fong, Y. Enhancement of oncolytic properties of recombinant newcastle disease virus through antagonism of cellular innate immune responses. Mol. Ther. 2009, 17, 697–706, doi:10.1038/mt.2008.286.
Krug, R.M. Functions of the influenza A virus NS1 protein in antiviral defense. Curr. Opin. Virol. 2015, 12, 1–6, doi:10.1016/j.coviro.2015.01.007.
- Methods: please provide more details about the media, supplements, source used to grow primary GBM cells
We really appreciate this observation. We have added the details that he/she has suggested, adding the following statement in Material and Methods section:
“All GBM cells were obtained and grown in M21 media containing containing DMEM/F-12 (Gibco, 11039), Non Essential Amino Acids (10mM; Gibco, 11140), Hepes (1M; Gibco, 15630), D-Glucose (45%; Sigma, G8769), BSA-F5 (7,5%; Gibco, 15260), Sodium Pyruvate (100mM; Gibco, 11360), L-Glutamine (200mM; Gibco, 25030), Antibiotic- Antimycotic (100x; Gibco, 15240), N2 Supplement (100x;Gibco, 17502), Hydrocortisone (1μg/μl; Sigma, H0135), Tri-iodothyronine (100μg/ml; Sigma, T5516), EGF (25ng/ μl; Sigma, E9644), bFGF (25ng/μl; Sigma, F0291) and Heparin (1μg/μl; Sigma, H3393), following the same process previously described [26]. Cells were maintained at 37 οC with 5% of CO2 and 90% humidity incubator. Lipofectamine™ 3000 Reagent (Thermofisher Scientific) was used to transfect CSCs with luciferase-expressing plasmids (pLV[Exp]-Hygro-EF1A>Luciferase) following manufacter’s procedure.”
- When appropriate, please include proper statistical analysis in figure 1 (for ex. Fig. 1e).
Although we understand the concern of Reviewer #3, we would invite him/her to review the rest of the experiments that we have done. We have performed the statistical analysis using a 2-tailed Student t test and two-way ANOVA in all the figures shown, except for the Figure 1, since the main aim of the analysis was to study the prevalence of CDKN2A and IFN gene aberration in glioma patients and not to compare among them.
However, if the Reviewer#3 still considers that a statistical analysis should be included in the figure 1, we will be willing to follow his directions.
- Figure 5b right panel is mislabeled, not clear which one shows an MOI of 1 since both are labeled MOI 0.01.
We really appreciate and thank the Reviewer #3 for pointing out this mistake, which is also a concern for Reviewer #1 and #2. We have labelled it correctly.
- There are numerous errors in the text, for ex:
Line 79 : “dead” should be cell death ? Line 115: “GMB” should be GBM
Line 199 : “shown” should be shows, Lines 252, 253, etc.
We thank the Reviewer #3 for these comments. We have reviewed and corrected these sentences and other grammatical typos throughout the manuscript.
We look forward to hearing from you regarding our submission. We would be glad to respond to any further comments that you may have.
Round 2
Reviewer 2 Report
1) "For this reason, the first aim of our study was an in silico assay including more than 1000 patients. Once we focused our hypothesis, we analyzed 6 GBM CSCs and then decided to continue with one representative cell line of each group. As we have continued with in vitro, in vivo and patients’ studies, we believe that our study is well-founded. Moreover, the CSCs selected (GBM18 and GBM27) have been previously characterized by our group in the context of their original tumors and the in vitro and in vivo evolution. However, if the Reviewer#2 still considers that we should add more cell lines, we will be willing to follow his directions."
Though I appreciate the need to make cost effective decisions when planning experiments, the choice of GBM27 as the representative cell line still strikes me as poor, given the unique nature of this sample in comparison to the two others in the same subgroup. More explanation must be provided for this choice (beyond having previously characterized its in vitro and in vivo evolution) when choosing it as a representative example. It hampers the conclusions one can draw given its distinctive character.
2) With regards to the growth kinetics, I was referring to Fig. 2a and not cell viability.
Author Response
Reviewer #2:
1) Though I appreciate the need to make cost effective decisions when planning experiments, the choice of GBM27 as the representative cell line still strikes me as poor, given the unique nature of this sample in comparison to the two others in the same subgroup. More explanation must be provided for this choice (beyond having previously characterized its in vitro and in vivo evolution) when choosing it as a representative example. It hampers the conclusions one can draw given its distinctive character.
To better explain why we have chosen GBM27 as a representative model of IFN deficient CSCs, we would like to invite the Reviewer#2 to see the Figure 3b. When we infected the six CSCs with NDV at different MOIs, the GBM27 was the more sensitive to the viral infection, showing a reduction in cell viability lower than 50%, with a very low MOI (0.001 Moreover, in Fig 2a GBM128B and GBM128D present a similar pattern as compared to GBM27, although we observed a slight induction of MX1 at 96h, for these reasons we thought that GBM27 was the most clean phenotype for CSCs that represents non-competent GBM model. As indicated in the manuscript, it is possible that GBM128B and GBM128D may contain some level of heterogeneity.
As we would like to better clarify this point to the Reviewer, we have checked the effect of NDV-NS1 in GBM38 another competent CSC model and in GBM128B as IFN-deficient CSC and added to Supplementary section (Figure S6).
Figure S6. NDV-NS1 restored oncolytic activity in competent CSCs model. GBM38 and GBM128D were exposed to NDV-GFP and NDV-NS1 MOI 1 and 0.01. Cell viability was measured every 24 h during 120 h. Error bar represents Standard Deviation. *P<0.05, **P<0.01, ***P<0.001.
We have added new information in the main text, in the results section page 12, lines 338-345: “For this purpose, we used GBM18 and GBM38 as IFN competent CSCs and GBM27 and GBM128D as deficient CSCs. As shown in Figure 5a, rNDV-NS1 viral titer was 1000-fold higher than rNDV-GFP virus in GBM18 and were more effective at high MOI reducing the number of viable cells below 20% at 120 h. The rNDV-NS1 decreased the viability in GBM38 drastically compared with rNDV-GFP at both MOIs. However, the growth kinetics of rNDV-GFP and rNDV-NS1 viruses did not change in GBM27 and GBM128D and similar viability effects were observed when these CSCs were infected with MOI 0.01 and 1 (Figure 5b and Figure S6)”.
We believe that these current data re-confirm our conclusions and improved the manuscript.
2) Is there any discernable difference in growth kinetics over a range of MOIs (rather than MOI=0.01, which was reported)?
With regards to the growth kinetics, I was referring to Fig. 2a and not cell viability.
Sorry for the confusion.
In general, the induction of host responses depends of the multiplicity of infection (MOI), which is related to the number of virus per cell. When we used high MOI values, most of the CSCs are infected rapidly and most of them start dying, so is not possible to study the growth kinetics for 120 h. As the reviewer knows, viruses cannot replicate without the machinery of host cells. So, in the field of virology the use of low MOIs to study the growth kinetic is a well-established practice. CSCs treatment with low MOIs results in a slower progress of infection that allows us to study the NDV growth curve shown in Figure 2b.
In this sense, we, and others, have infected several cell lines (CEFs, Vero, Hep-2, and A549) at MOI of 0.01 to study NDV replication curve in a previous publication in the Journal of Virology.
References:
-Fields of Virology Chapter 2: Editors David Mahan Knipe, Peter M. Howley.
-New advances on Zika Virus Research: Editors Luis Martínez-Sobrido, Fernando Almazan Toral.
-Park, M.-S.; Garcia-Sastre, A.; Cros, J.F.; Basler, C.F.; Palese, P. Newcastle Disease Virus V Protein Is a Determinant of Host Range Restriction. J. Virol. 2003, 77, 9522–9532, doi:10.1128/jvi.77.17.9522-9532.2003.

Reviewer 3 Report
The authors have responded to the critiques and have provided satisfactory answers. I believe that this revised manuscript has been improved and is acceptable for publication.
Author Response
We thank the reviewer 3 for his/her comments
Round 3
Reviewer 2 Report
The authors have sufficiently responded to my concerns.